# Responses of Tobacco Growth and Development, Nitrogen Use Efficiency, Crop Yield and Economic Benefits to Smash Ridge Tillage and Nitrogen Reduction

**Bufan Zheng** [1], **Yongfeng Jing** [2], **Yidong Zou** [1], **Ruiwen Hu** [1], **Yongjun Liu** [3], **Zhipeng Xiao** [4], **Fei He** [5], **Qiyun Zhou** [1], **Xiangshen Tian** [1], **Jia Gong** [1], **Juan Li** [1] **and Zhongwen Rang** [1,*]

1 Agronomy College, Hunan Agricultural University, Changsha 410128, China
2 Hunan Tobacco Industry Co., Ltd., Changsha 410014, China
3 Hunan Tobacco Science Institute, Changsha 410004, China
4 Hunan Tobacco Company Hengyang Branch, Hengyang 421000, China
5 Hunan Tobacco Company Changde Branch, Changde 415000, China
* Correspondence: rangzhongwen@hunau.edu.cn

**Abstract:** Long-term rotary tillage leads to thinning of the soil layer and low nitrogen use efficiency of crops, resulting in a decrease in crop yield and quality. Therefore, a new alternative method to improve nitrogen use efficiency of crops needs to be found urgently. Here, we analyzed the effects of a new smash ridge tillage method combined with reduced nitrogen application on tobacco growth and development, and nitrogen use efficiency and its economic benefits. The results showed that, compared with conventional tillage and nitrogen application of 180 kg N ha$^{-1}$, smash ridge tillage and a 30% reduction in nitrogen rate resulted in greater root length density, more primary lateral roots and greater rooting depth in the subsoil. It is also beneficial to maintain a high level of biomass and nitrogen accumulation in the later growth period, increasing the output value of tobacco by CNY 1588.35 ha$^{-1}$ and reducing the cost by CNY 974.1 ha$^{-1}$ on average in two years. In conclusion, our study highlights the economic benefits of smash ridge tillage and nitrogen reduction for tobacco growth and development, and considers them an effective method for improving agricultural productivity and nitrogen use efficiency.

**Keywords:** smash ridge tillage; nitrogen reduction; crop growth; yield; net economic benefit



## 1. Introduction

Soil is the foundation of agriculture and an indispensable important resource for social development [1]; good soil quality is the key factor for high-quality crop production and sustainable agriculture [2]. However, traditional farming methods and excessive use of chemical fertilizers are seriously damaging the soil quality of cultivated land [3], resulting in soil compaction [4], enhanced root resistance and deterioration of soil physical properties (such as soil aeration and water content) in many areas [5]. Appropriate crop planting patterns and effective nutrient management strategies are the key factors for promoting crop growth and yield, and sustainable crop yield plays an important role in the economy [6]. For a long time, tobacco–rice continuous cropping rotation fields in Hunan Province [7] have been dominated by traditional rotary tillage. The hazards of long-term rotary tillage to cultivated soil are as follows: the tillage layer becomes thinner (even less than 15 cm), the plow bottom moves upward and the soil permeability and soil compaction deteriorate [8]. In addition, the inhibition of soil mineralization leads to the slow decomposition and release of soil nutrients, which makes it difficult to achieve efficient absorption and utilization of crop growth [9], thus affecting the yield and quality of the crop [10].

Deep tillage is considered an effective way to improve crop yield [9]. Holland [11] pointed out that, compared with rotary tillage and harrowing tillage, deep tillage has a

positive effect on improving soil carbon sequestration in farmland. Deep tillage combined with straw returning can improve soil structure and affect microbial communities [12], help crops resist low temperature to promote seedling emergence [13], promote the diffusion of crop roots and improve crop nutrient absorption and yield [14].

The smash ridge tillage mentioned in this experiment uses the special high-speed rotary drill of a smash ridge tillage machine to crush the soil vertically above 30 cm (up to 80 cm) and loosen the soil further based on a conventional tillage layer (15~20 cm). It can deepen the tillage depth, increase soil water retention capacity, improve soil physical properties, stimulate soil potential nutrients [15], create a more suitable growth environment for plant roots, enable plants to better absorb soil nutrients and is conducive to the growth and development of crops [16]. Kahlon and Khurana [17] found that deep tillage increased root length density of maize and wheat by 44% and 34%, respectively, compared with conventional tillage, and the yields of maize and wheat under deep tillage were 14% and 12% higher than those under conventional tillage, respectively. At the same time, the application of deep tillage technology in legumes [18], watermelon [19], rice [20], tea [21] and other crops was also studied. Compared with conventional tillage, crop yield and quality and economic benefits were improved to varying degrees.

Tobacco is a crop that requires a large amount of nitrogen, and the average nitrogen application rate is 180 kg ha$^{-1}$. To improve crop growth and production, farmers have increased the rate of nitrogen fertilizer application in common tillage farming systems in China, which is not environmentally eco-friendly [22,23]. Wang et al. [24] pointed out that the application of more than 240 kg N ha$^{-1}$ could promote the downward movement of N-15 and soil nitrate at the base and top of wheat plants, but had no significant effect on the amount of nitrogen absorption. Cameron et al. [25] also pointed out that excessive application of chemical fertilizers can significantly reduce the nitrogen use efficiency of crops, and cause serious nitrogen loss, which is not conducive to reducing cultivated land pollution and ensuring farmers' income. Currently, most studies on improving nitrogen use efficiency focus on fertilization methods and types of fertilizers [26], but there are few reports on the combination of fertilization methods with tillage methods.

Therefore, in order to solve the problems of continuous cropping obstacles, low fertilizer utilization rate and unsatisfactory crop yield and quality in Hunan Province, we used tobacco as the model crop. Based on the assumption that smash ridge tillage can increase yield compared with conventional tillage [27], combined with reducing the amount of nitrogen fertilizer, the effect of reducing nitrogen fertilizer after smash ridge tillage on tobacco growth and economic traits was explored, and the reasonable nitrogen reduction interval after smash ridge tillage was clarified. It provides a reference for the application of smash ridge tillage technology in crop production and agricultural sustainable development in the future.

## 2. Materials and Methods

### 2.1. Experimental Site and Variety Choice

Before the flue-cured tobacco planting seasons in 2018 and 2019, a tobacco–rice compound cropping field with flat terrain and uniform fertility was selected in Daozi Village, Daozi Township, Leiyang City for smash-ridging tillage, and nitrogen reduction treatment was carried out during the flue-cured tobacco planting season. The local area has a humid subtropical monsoon climate, and the weather conditions during the flue-cured tobacco planting season are shown in Table 1. There was no extreme weather during the 2-year test period. The soil type of the test site is paddy soil, and the soil texture is sandy loam (clay 11.4%, silt 24.9%, sand 63.6% and gravel 0.1%). The depth of the plow bottom is about 12 cm. The soil fertility status is shown in Table 2. The sampling time is after the late rice harvest, and the field had been drained. The tested variety was Yunyan 87. The method of raising tobacco seedlings is carried out in floating trays. These are perforated foam boards, and 3–5 tobacco seeds were sown in each hole, and the sowing time was around 15 January, in a greenhouse. The transplanting time was from 17 to 22 March of the two years (when it

did not rain), and other field management measures were based on the local high-quality tobacco leaf production technical manual.

**Table 1.** Weather conditions of flue-cured tobacco planting season.

|  | Average Maximum Temperature/°C | Average Minimum Temperature/°C | Average Rainfall/mm |
|---|---|---|---|
| March | 24 | 17 | 195.0 |
| April | 28 | 21 | 232.5 |
| May | 30 | 24 | 270.0 |
| June | 32 | 26 | 204.6 |
| July | 32 | 26 | 229.4 |

**Table 2.** Soil fertility tested in two years.

| Year | Soil Depth (cm) | pH | Soil Organic Carbon (g kg$^{-1}$) | Total N (g kg$^{-1}$) | Total P (g kg$^{-1}$) | Total K (g kg$^{-1}$) | Alkaline N (mg kg$^{-1}$) | Available P (mg kg$^{-1}$) | Available K (mg kg$^{-1}$) |
|---|---|---|---|---|---|---|---|---|---|
| 2018 | 0~10 | 6.9 | 55.51 | 2.39 | 1.00 | 16.33 | 197.12 | 31.75 | 113.23 |
|  | 10~20 | 6.8 | 52.37 | 2.30 | 0.90 | 15.58 | 193.29 | 25.24 | 67.45 |
|  | 20~30 | 7.3 | 50.79 | 2.22 | 0.69 | 14.48 | 172.24 | 10.13 | 57.26 |
| 2019 | 0~10 | 6.8 | 49.00 | 3.34 | 0.85 | 20.60 | 172.12 | 23.00 | 255.25 |
|  | 10~20 | 7.0 | 47.30 | 4.01 | 0.89 | 21.20 | 191.24 | 22.00 | 300.03 |
|  | 20~30 | 6.8 | 44.40 | 2.32 | 0.80 | 21.00 | 159.56 | 4.00 | 245.03 |

*2.2. Experimental Machine*

The smash-ridging tillage machine manufactured by Guangxi Wu Feng Machinery Co., Ltd., Yulin, China (Figure 1) is equipped with 60 cm long spiral blade drill pipes, the spacing of the rod shank is 36 cm (between centers) and the engine power is 239 kW. The smash ridge tillage machine in this study was developed by Hunan Agricultural University and Guangxi Wu Feng Machinery Co., Ltd., and the code is SGL-160.

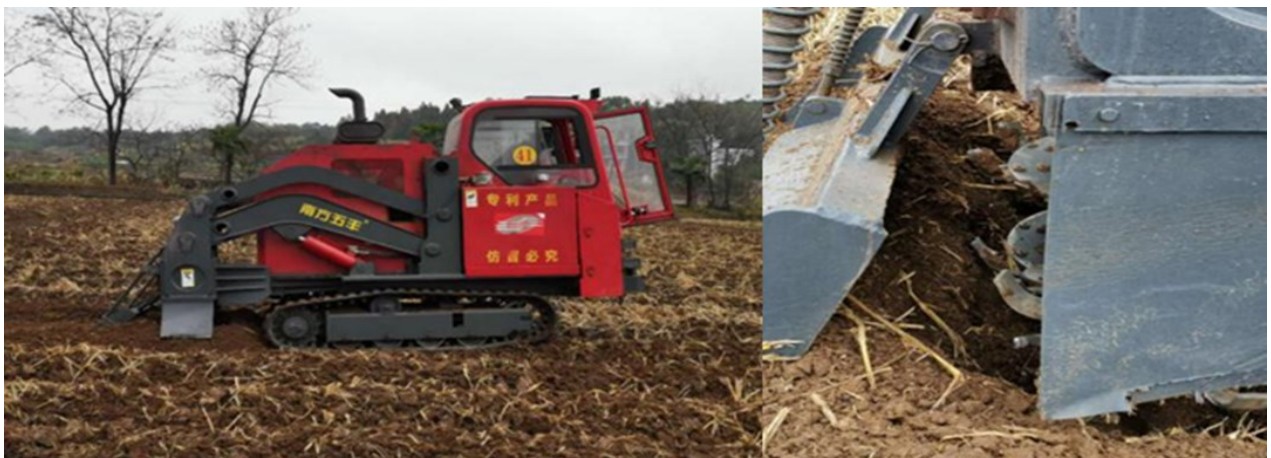

**Figure 1.** The smash ridge tillage machine.

*2.3. Experimental Design*

The experiment adopted a randomized block design, and in 2018, two treatments with reduced nitrogen fertilizer application were used: smash ridge tillage + 15% nitrogen fertilizer reduction (N85), and smash ridge tillage + 30% nitrogen fertilizer reduction (N70). There were also two control treatments: smash ridge tillage + normal nitrogen application (N100), and conventional tillage + normal nitrogen application (CK). In order to further explore the threshold value of nitrogen fertilizer application reduction and

optimize the ratio of nitrogen fertilizer application reduction, in 2019, four treatments of nitrogen fertilizer reduction were set up in another experimental field: smash ridge tillage + 10% reduction in nitrogen fertilizer application (N90), smash ridge tillage + 20% nitrogen fertilizer reduction (N80), smash ridge tillage + 30% nitrogen fertilizer reduction (N70) and smash ridge tillage + 40% nitrogen fertilizer reduction (N60). There were also 2 control treatments: smash ridge tillage + normal nitrogen fertilization (N100), and conventional tillage + normal nitrogen fertilization (CK). In the experiments in 2018 and 2019, three plots were arranged for all treatments, the plot area was about 60 m$^2$ and the row-to-plant spacing of flue-cured tobacco was 1.2 m $\times$ 0.5 m (0.6 m$^2$). The nitrogen reduction scheme was implemented when the ground fertilizer was applied, and the amount of base fertilizer was converted according to the total amount of nitrogen fertilizer in each treatment. The fertilizer method is a hole application before the transplanting, and the missing P$_2$O$_5$ and K$_2$SO$_4$ were supplemented with nitrogen-free fertilizer. Top dressing was applied three times before topping, the application method was irrigation and the amount of each treatment was the same. The prices of the ground fertilizer, potassium sulfate and calcium superphosphate were CNY 2.43 kg$^{-1}$, CNY 2.54 kg$^{-1}$ and CNY 0.4 kg$^{-1}$, respectively. The various fertilizing amount is shown in Table 3. The equations for application rate are as follows:

$$Application\ rate = the\ amount\ per\ hectare\ /\ the\ number\ of\ plants\ per\ hectare, \quad (1)$$

$$Number\ of\ plants\ per\ hectare = 1\ hectare\ /\ the\ unit\ area\ of\ each\ plant. \quad (2)$$

**Table 3.** Conversion of fertilizer consumption.

| Year | Treatment/Fertilizer Type | Base Fertilizer (kg ha$^{-1}$) | Calcium Superphosphate (kg ha$^{-1}$) | Potassium Sulfate (kg ha$^{-1}$) | N (kg ha$^{-1}$) |
|------|---------------------------|-------------------------------|---------------------------------------|----------------------------------|------------------|
| 2018 | N100 | 2250 | 0 | 0 | 180 |
|      | N85 | 1912.5 | 309.4 | 74.3 | 153 |
|      | N70 | 1575 | 511.4 | 122.7 | 126 |
|      | CK | 2250 | 0 | 0 | 180 |
| 2019 | N100 | 2250 | 0 | 0 | 180 |
|      | N90 | 2025 | 206.3 | 49.5 | 162 |
|      | N80 | 1800 | 412.5 | 99.0 | 144 |
|      | N70 | 1575 | 511.4 | 122.7 | 126 |
|      | N60 | 1350 | 815 | 198 | 108 |
|      | CK | 2250 | 0 | 0 | 180 |

*2.4. Test Items and Methods*

Complete plant roots (5 representative plants per plot) were collected 60 and 80 days after transplanting and soaked in clear water and washed, and the soil adhering to the root was picked off with a mesh sieve. The first lateral root was measured by manual statistics, the root volume was measured by water immersion method and the root depth was measured by tape ruler. Then, the agronomic traits were determined [28]. Three tobacco plants were sampled at 60 d and 80 d after transplanting, respectively. The effective leaf number was measured by manual statistics, and maximum leaf length and maximum leaf width were measured by tape ruler. The equation for maximum leaf area is as follows:

$$Maximum\ Leaf\ area = leaf\ length * leaf\ width * 0.6345. \quad (3)$$

Then, plant samples were investigated. Three tobacco plants were sampled at 60 d and 80 d after transplanting, and then partially dried and weighed. The plant nitrogen was determined by the Kjeldahl method. The equations are as follows [29]:

$$
\begin{aligned}
&Nitrogen\ accumulation\ (g/plant) = \\
&dry\ matter\ mass\ (g)\ of\ one\ organ\ of\ tobacco\ plant * \\
&nitrogen\ con-tent(\%)\ of\ one\ organ\ of\ tobacco\ plant,
\end{aligned} \tag{4}
$$

$$
\begin{aligned}
&Nitrogen\ dry\ matter\ production\ efficiency \\
&= dry\ weight\ per\ plant/N\ uptake\ per\ plant,
\end{aligned} \tag{5}
$$

$$
\begin{aligned}
&Nitrogen\ harvest\ index \\
&= leaf\ nitrogen\ accumulation/plant\ nitrogen\ accumulation.
\end{aligned} \tag{6}
$$

Then, the economic characters were counted. In each plot, 30 tobacco plants were selected for listing and baking, and the yield was calculated separately. The yield, output value, proportion of medium tobacco and proportion of superior tobacco were converted according to the area occupied by the selected tobacco plants. The calculation method is as follows: Economic benefit = Total output value − Total fertilizer cost − Total labor (constant excluded).

*2.5. Statistical Analysis*

Excel 2019 was used for data statistics and tabulation. SPSS 22.0 (SPSS Inc., Chicago, IL, USA) was used to identify the significant differences between the treatments by using analysis of variance (ANOVA) and least significant difference (LSD) approaches. Differences were considered statistically significant when $p < 0.05$. Duncan's new complex range method was used to test the differences. The figures were made using Sigma Plot 12.0 (Systat Software, Inc., San Jose, CA, USA).

## 3. Results

*3.1. Agronomic Characteristics and Dry Matter Quality*

Table 4 shows that, in the 2018 experiments, the maximum leaf area at 60 d after transplanting of each treatment was significantly larger than that of CK, and N70 showed the best agronomic traits at 80 days after transplantation. The maximum leaf area was 78.91 cm$^2$ larger than that of CK. In terms of root dry matter accumulation, N100, N85 and N70 root dry matter increased by 1.05 g, 1.49 g and 6.09 g, respectively, compared with CK. N100 had the highest leaf dry matter accumulation, 4.79 g higher than CK. There was no significant difference in dry matter accumulation between N85 and N70. In the 2019 experiment, 60 d after transplanting, all indexes of N60 were significantly lower than those of other treatments; there was no significant difference in agronomic traits among N100, N90, N80 and N70 at 80 d after transplanting, and the indexes of N60 were significantly lower than those of other treatments. The dry matter accumulation of N80 was the highest, and the dry matter accumulation of N100 and N70 was slightly lower than that of N80 but significantly higher than that of CK. No significant difference in dry matter accumulation between N70 and N100 was found. The dry matter accumulation in all the parts of N60 was significantly lower than that in other treatments. The leaf number per plant of N70 was the highest.

**Table 4.** Effects of reducing nitrogen application on agronomic traits.

| Year | Sampling Time (after Transplanting) | Treatment | Blade Area (max)/cm² | Leaf Number/Plant | Dry Matter Weight | | |
|---|---|---|---|---|---|---|---|
| | | | | | Root/g | Stem/g | Leaf/g |
| 2018 | 60 d | N100 | 1334.18 ± 28.93 [a] | 18.0 ± 0.0 [b] | 75.44 ± 2.27 [b] | 67.57 ± 1.60 [b] | 123.97 ± 1.46 [b] |
| | | N85 | 1297.53 ± 78.80 [a] | 17.3 ± 0.6 [c] | 75.44 ± 2.27 [b] | 65.39 ± 0.44 [c] | 118.53 ± 0.68 [d] |
| | | N70 | 1321.70 ± 38.91 [a] | 19.0 ± 0.0 [a] | 79.53 ± 0.52 [a] | 66.56 ± 0.87 [bc] | 120.92 ± 1.07 [c] |
| | | CK | 1283.28 ± 32.35 [a] | 19.0 ± 0.0 [a] | 78.80 ± 1.85 [ab] | 70.78 ± 0.53 [a] | 126.88 ± 0.80 [a] |
| | 80 d | N100 | 1332.61 ± 81.51 [a] | 18.0 ± 1.0 [a] | 81.95 ± 0.92 [b] | 88.37 ± 0.81 [a] | 130.38 ± 0.67 [a] |
| | | N85 | 1317.79 ± 33.72 [a] | 18.7 ± 1.5 [a] | 82.95 ± 1.13 [b] | 88.65 ± 0.31 [a] | 128.30 ± 0.56 [bc] |
| | | N70 | 1329.38 ± 18.24 [a] | 18.7 ± 1.2 [a] | 86.99 ± 1.60 [a] | 87.97 ± 0.74 [a] | 128.82 ± 0.30 [b] |
| | | CK | 1250.47 ± 48.48 [a] | 17.3 ± 2.1 [a] | 80.90 ± 0.54 [b] | 83.86 ± 0.22 [b] | 127.22 ± 1.05 [c] |
| 2019 | 60 d | N100 | 1346.84 ± 96.89 [a] | 18.2 ± 0.5 [a] | 75.06 ± 0.33 [b] | 77.26 ± 0.25 [a] | 122.23 ± 0.41 [a] |
| | | N90 | 1312.47 ± 21.91 [a] | 17.3 ± 0.6 [ab] | 75.50 ± 0.64 [b] | 74.36 ± 0.94 [bc] | 117.23 ± 0.91 [c] |
| | | N80 | 1338.06 ± 77.52 [a] | 18.2 ± 1.1 [a] | 77.48 ± 1.09 [a] | 74.36 ± 1.86 [bc] | 120.33 ± 0.90 [b] |
| | | N70 | 1343.68 ± 110.09 [a] | 17.0 ± 0.0 [b] | 76.20 ± 0.80 [ab] | 76.60 ± 1.37 [ab] | 120.10 ± 1.35 [b] |
| | | N60 | 722.40 ± 30.75 [b] | 15.4 ± 0.5 [c] | 56.03 ± 1.48 [d] | 51.93 ± 1.39 [d] | 60.63 ± 1.33 [d] |
| | | CK | 1199.21 ± 90.31 [a] | 17.7 ± 0.6 [ab] | 67.26 ± 0.85 [c] | 72.48 ± 1.03 [c] | 119.05 ± 0.97 [bc] |
| | 80 d | N100 | 1315.31 ± 58.37 [a] | 18.0 ± 0.0 [a] | 84.96 ± 0.59 [a] | 79.33 ± 0.30 [a] | 154.76 ± 1.01 [a] |
| | | N90 | 1309.41 ± 43.61 [a] | 18.8 ± 0.2 [a] | 83.63 ± 0.60 [a] | 75.03 ± 1.19 [b] | 150.93 ± 1.67 [b] |
| | | N80 | 1314.46 ± 65.14 [a] | 18.3 ± 1.2 [a] | 85.76 ± 1.72 [a] | 80.16 ± 1.12 [a] | 155.96 ± 1.42 [a] |
| | | N70 | 1347.98 ± 42.24 [a] | 18.7 ± 0.6 [a] | 84.66 ± 2.19 [a] | 80.03 ± 1.56 [a] | 153.33 ± 1.48 [ab] |
| | | N60 | 829.28 ± 27.97 [b] | 16.0 ± 0.0 [b] | 63.73 ± 1.03 [c] | 54.00 ± 1.14 [d] | 66.70 ± 1.02 [d] |
| | | CK | 1266.79 ± 68.62 [a] | 17.7 ± 0.6 [a] | 80.31 ± 1.11 [b] | 72.51 ± 1.81 [c] | 144.62 ± 1.84 [c] |

Note: Lowercase letters are significantly different at the 0.05 level, maximum leaf means middle leaf (7th or 8th from top to bottom).

*3.2. Root Growth*

As Table 5 and Figure 2 shows, 60 d and 80 d after transplanting in 2018, the average root depth of DT treatments was 24.68 cm and 26.54 cm, respectively, which was 4.61 cm and 4.04 cm higher than that of CK, and the root setting depth of the N70 treatment was the largest, reaching 26.27 cm and 28.17 cm at 60 d and 80 d after transplanting, respectively. In terms of the increase in lateral root number (equal to the side root number 80 d after transplanting minus the side root number 60 d after transplanting), the increase in lateral root number of N70 was greater than that of CK. In the 2019 experiments, 60 d after transplanting, the rooting depth of N90 and N80 was significantly deeper than that of N100, and compared with N100, N80 and N70, significantly increased the volume by 61.66 cm$^2$ and 43.40 cm$^2$, respectively. At 80 d after transplanting, the volume of lateral roots in the treatment of nitrogen reduction after DT was significantly higher than that of N100, with the largest volume in N80; from 60 d to 80 d after transplanting, the growth rate of rooting depth of N90, N80 and N70 was lower than that of N100, but the growth rate of lateral root volume was higher than that of N100.

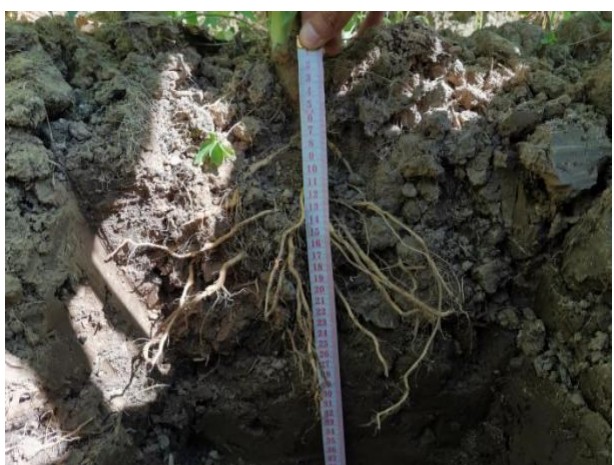 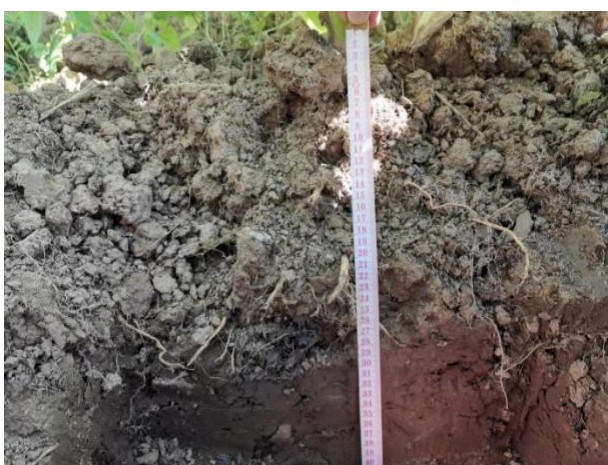

**Figure 2.** Comparison of root systems between DT and CK. Note: The left was for DT, and the right was for conventional tillage.

**Table 5.** Influence of reduced nitrogen application on root depth and lateral root number.

| Year | Sampling Time (after Transplanting) | Treatment | Root Depth/cm | Number of Lateral Roots | Volume/cm$^2$ |
|------|------|------|------|------|------|
| 2018 | 60 d | N100 | 24.20 ± 0.95 [b] | 16.67 ± 1.15 [b] | — |
| | | N85 | 23.57 ± 1.10 [b] | 20.00 ± 3.00 [ab] | |
| | | N70 | 26.27 ± 0.78 [a] | 18.33 ± 2.08 [b] | |
| | | CK | 20.07 ± 0.25 [c] | 22.67 ± 2.08 [a] | |
| | 80 d | N100 | 25.93 ± 0.76 [b] | 21.67 ± 1.15 [a] | |
| | | N85 | 25.53 ± 1.16 [b] | 22.00 ± 3.46 [a] | |
| | | N70 | 28.17 ± 1.10 [a] | 22.67 ± 2.31 [a] | |
| | | CK | 22.50 ± 1.25 [c] | 25.67 ± 0.58 [a] | |

**Table 5.** *Cont.*

| Year | Sampling Time (after Transplanting) | Treatment | Root Depth/cm | Number of Lateral Roots | Volume/cm$^2$ |
|---|---|---|---|---|---|
| 2019 | 60 d | N100 | 20.73 ± 0.60 [c] | 20.00 ± 4.00 [ab] | 86.67 ± 2.44 [d] |
| | | N90 | 25.73 ± 1.20 [a] | 21.33 ± 0.58 [ab] | 93.33 ± 0.97 [c] |
| | | N80 | 22.73 ± 0.22 [b] | 22.33 ± 0.58 [a] | 148.33 ± 0.67 [a] |
| | | N70 | 20.73 ± 0.25 [c] | 22.00 ± 0.00 [a] | 130.07 ± 7.90 [b] |
| | | N60 | 16.33 ± 0.47 [e] | 15.67 ± 0.58 [b] | 65.07 ± 0.49 [e] |
| | | CK | 18.07 ± 1.72 [d] | 18.00 ± 2.00 [bc] | 84.00 ± 1.45 [d] |
| | 80 d | N100 | 22.67 ± 0.67 [a] | 23.67 ± 0.58 [c] | 115.03 ± 2.76 [d] |
| | | N90 | 23.97 ± 0.78 [a] | 29.33 ± 0.58 [b] | 142.03 ± 1.69 [c] |
| | | N80 | 24.33 ± 1.86 [a] | 30.00 ± 0.00 [b] | 161.73 ± 1.86 [a] |
| | | N70 | 24.33 ± 1.64 [a] | 32.67 ± 0.58 [a] | 151.70 ± 1.91 [b] |
| | | N60 | 17.33 ± 0.32 [c] | 21.00 ± 0.00 [d] | 110.00 ± 1.73 [e] |
| | | CK | 20.00 ± 0.56 [b] | 20.67 ± 0.58 [d] | 104.03 ± 3.43 [f] |

Note: Lowercase letters are significantly different at the 0.05 level.

*3.3. Nitrogen Absorption*

It can be seen from Table 6 that the effect of smash ridge tillage combined with reducing nitrogen application on crop nitrogen uptake is reflected in the middle and late stages of crop growth. In the 2018 experiment, the N uptake per plant and N accumulation in tobacco leaves of each treatment were significantly lower than those of CK 60 d after transplanting. However, the situation reversed at 80 d after transplanting. The nitrogen uptake per plant and nitrogen accumulation in tobacco leaves of each treatment in smash ridge tillage were significantly higher than those of CK, and the highest was N70. The nitrogen uptake per plant and nitrogen accumulation in tobacco leaves were 9.12 g and 3.08 g plant$^{-1}$, respectively. The nitrogen harvest index of DT was higher than that of CK. In the 2019 experiment, after 60 days of transplanting, there was no significant difference in nitrogen uptake per plant and nitrogen accumulation in tobacco leaves among treatments except N60. Eighty days after transplanting, N90 was significantly higher than other treatments in terms of nitrogen uptake per plant, and there was no significant difference among N70, N80 and N100. N70, N80 and N90 had a higher nitrogen harvest index than CK.

**Table 6.** Effect of reduced nitrogen application on nitrogen uptake in roasted tobacco.

| Year | Sampling Time (after Transplanting) | Treatment | N Accumulation (g Plant$^{-1}$) | N Accumulation (g Leaf$^{-1}$) | N Production Efficiency | N Harvest Index (%) |
|---|---|---|---|---|---|---|
| 2018 | 60 d | N100 | 3.54 ± 0.34 [c] | 1.46 ± 0.26 [b] | 48.19 ± 7.08 [a] | 82.49 ± 3.84 [a] |
| | | N85 | 3.49 ± 0.19 [c] | 1.40 ± 0.17 [b] | 47.24 ± 3.55 [a] | 77.65 ± 3.63 [ab] |
| | | N70 | 4.90 ± 0.13 [b] | 1.80 ± 0.05 [b] | 45.26 ± 1.24 [a] | 64.90 ± 2.45 [b] |
| | | CK | 6.41 ± 0.42 [a] | 3.52 ± 0.39 [a] | 43.13 ± 2.88 [b] | 56.94 ± 3.16 [c] |
| | 80 d | N100 | 8.35 ± 0.17 [b] | 2.72 ± 0.08 [b] | 40.91 ± 0.60 [a] | 45.31 ± 1.55 [a] |
| | | N85 | 8.46 ± 0.21 [b] | 2.85 ± 0.21 [b] | 40.13 ± 0.80 [a] | 46.28 ± 2.71 [a] |
| | | N70 | 9.12 ± 0.19 [a] | 3.08 ± 0.04 [a] | 37.41 ± 0.47 [b] | 36.73 ± 1.48 [b] |
| | | CK | 7.14 ± 0.26 [c] | 2.69 ± 0.05 [b] | 40.89 ± 1.30 [a] | 33.42 ± 2.82 [b] |

**Table 6.** *Cont.*

| Year | Sampling Time (after Transplanting) | Treatment | N Accumulation (g Plant$^{-1}$) | N Accumulation (g Leaf$^{-1}$) | N Production Efficiency | N Harvest Index (%) |
|---|---|---|---|---|---|---|
| 2019 | 60 d | N100 | 5.71 ± 0.42 [a] | 2.97 ± 0.49 [a] | 48.08 ± 5.15 [a] | 71.24 ± 3.60 [a] |
| | | N90 | 6.15 ± 0.29 [a] | 3.15 ± 0.19 [a] | 48.43 ± 0.68 [a] | 66.87 ± 2.17 [a] |
| | | N80 | 5.51 ± 0.44 [a] | 2.65 ± 0.27 [a] | 49.62 ± 1.19 [a] | 68.61 ± 3.56 [a] |
| | | N70 | 5.63 ± 0.16 [a] | 2.78 ± 0.10 [a] | 48.47 ± 0.58 [a] | 67.96 ± 1.04 [a] |
| | | N60 | 3.94 ± 0.35 [b] | 1.80 ± 0.18 [b] | 42.79 ± 8.08 [b] | 47.11 ± 3.56 [b] |
| | | CK | 5.81 ± 0.65 [a] | 3.00 ± 0.41 [a] | 51.52 ± 2.45 [a] | 44.90 ± 5.16 [b] |
| | 80 d | N100 | 7.83 ± 0.35 [b] | 3.23 ± 0.18 [a] | 40.75 ± 1.97 [a] | 34.73 ± 0.83 [a] |
| | | N90 | 8.93 ± 0.26 [a] | 3.28 ± 0.06 [a] | 39.67 ± 1.05 [a] | 37.17 ± 1.28 [a] |
| | | N80 | 7.90 ± 0.30 [b] | 3.34 ± 0.17 [a] | 40.74 ± 1.78 [a] | 37.84 ± 0.49 [a] |
| | | N70 | 7.62 ± 0.50 [b] | 3.18 ± 0.20 [a] | 41.73 ± 2.95 [a] | 37.79 ± 1.72 [a] |
| | | N60 | 6.86 ± 0.31 [c] | 2.73 ± 0.23 [b] | 26.88 ± 0.86 [b] | 21.54 ± 1.60 [b] |
| | | CK | 6.96 ± 0.24 [c] | 2.58 ± 0.08 [b] | 42.75 ± 1.57 [a] | 37.10 ± 2.30 [a] |

Note: Lowercase letters are significantly different at the 0.05 level.

*3.4. Economic Traits*

Table 7 shows that, in the 2018 experiment, the yield of N100 was the highest, reaching 2326.33 kg/ha, but the proportion of superior tobacco was the lowest, which was 90.7%. The average price of tobacco was the lowest, which was CNY 21.97 kg$^{-1}$, and the total output value was CNY 51,070.37 ha$^{-1}$, which was lower than that of N70. The yield of N70 tobacco leaves was slightly lower than that of conventional tillage, but due to the proportion of upper and middle tobacco leaves and the average price of tobacco leaves being slightly higher, the output value of tobacco leaves was higher, which was CNY 51,780.4 ha$^{-1}$. In the experiment in 2019, the yield of N80 was the best, which was 440.10 kg ha$^{-1}$ higher than that of N100, but the proportion of upper and middle tobacco in N80 was lower than that of N70, so the output value of N70 was the highest, which was CNY 730.44 ha$^{-1}$ and CNY 2094.57 ha$^{-1}$ higher than that of N80 and N100, respectively, and the average price of N70 was the highest because of the highest output value and the proportion of upper and middle tobacco. The fertilizer cost is CNY 1124 ha$^{-1}$ lower than N100.

**Table 7.** Economic properties of tobacco leaves after curing.

| Year | Sampling Time (after Transplanting) | Yield (kg ha$^{-1}$) | Product Value (CNY ha$^{-1}$) | Mean Price (CNY ha$^{-1}$) | Ratio of Mid–High-Grade Leaves (%) |
|---|---|---|---|---|---|
| 2018 | N100 | 2326.33 ± 69.69 [a] | 51,070.37 ± 80.68 [b] | 21.97 ± 0.62 [a] | 90.7 ± 0.42 [c] |
| | N85 | 2299.07 ± 12.81 [a] | 51,080.60 ± 84.12 [b] | 22.22 ± 0.11 [a] | 92.0 ± 1.26 [ab] |
| | N70 | 2265.50 ± 28.78 [a] | 51,780.40 ± 369.97 [a] | 22.86 ± 0.15 [a] | 95.4 ± 0.52 [a] |
| | CK | 2304.17 ± 89.09 [a] | 50,925.17 ± 428.13 [b] | 22.12 ± 0.67 [a] | 93.8 ± 2.04 [a] |
| 2019 | N100 | 2335.17 ± 66.95 [b] | 51,132.60 ± 62.66 [d] | 21.90 ± 0.65 [bc] | 90.8 ± 1.07 [b] |
| | N90 | 2365.47 ± 31.98 [b] | 52,859.83 ± 242.95 [b] | 22.30 ± 0.21 [ab] | 90.4 ± 0.64 [b] |
| | N80 | 2475.27 ± 40.32 [a] | 52,496.73 ± 165.77 [c] | 21.20 ± 0.38 [c] | 90.0 ± 0.81 [b] |
| | N70 | 2320.57 ± 48.04 [bc] | 53,227.17 ± 169.25 [a] | 22.90 ± 0.40 [a] | 93.4 ± 0.51 [a] |
| | N60 | 1375.57 ± 33.39 [d] | 29,383.63 ± 317.81 [e] | 21.30 ± 0.29 [c] | 88.0 ± 3.23 [c] |
| | CK | 2248.33 ± 43.47 [c] | 50,905.70 ± 109.25 [d] | 22.60 ± 0.79 [a] | 91.4 ± 2.48 [ab] |

Note: Lowercase letters are significantly different at the 0.05 level.

*3.5. Net Economic Benefit*

As Table 8 shows, in the 2018 experiment, the highest total cost was N100 (CNY 8817.5 ha$^{-1}$), and the lowest was N70 (CNY 7693.5 ha$^{-1}$). The order of total cost from high to low was N100 > CK > N70, and the cost of CK was 11.24% higher than that of N70. The order of net income from high to low is N70 > CK > N100. Compared with N70, the net

income of CK is reduced by 4.33%, so the yield–input ratio of CK is lower than that of N70. In the smash ridge tillage treatments, the total cost of N100 was higher than that of N70, and the net income was lower than that of the nitrogen reduction treatment, so the yield–input ratio of N100 was lower than that of the nitrogen reduction treatment. In the experiment of 2019, the total cost increased gradually with the increase in nitrogen application rate, N70 had higher net income than N100, and N70 had the highest net income, which was 7.61% higher than N100.

**Table 8.** Effect of smash ridge tillage and reducing nitrogen fertilizer on economic benefit.

| Year | Treatment | Fertilizer Cost (CNY ha$^{-1}$) | Mechanical Costs (CNY ha$^{-1}$) | Total Cost (CNY ha$^{-1}$) | Net Income (CNY ha$^{-1}$) | Output Ratio (%) |
|------|-----------|----------------|-----------------|------------|------------|--------------|
| 2018 | N100 | 5467.5 | 3350.0 | 8817.5 | 42,252.87 ± 80.68 [b] | 20.87 ± 0.04 [a] |
|  | N70 | 4343.4 | 3350.0 | 7693.4 | 44,087.00 ± 369.97 [a] | 17.45 ± 0.14 [c] |
|  | CK | 5467.5 | 3200.0 | 8667.5 | 42,257.67 ± 428.13 [b] | 20.51 ± 0.21 [b] |
| 2019 | N100 | 5467.5 | 3350.0 | 8817.5 | 42,315.10 ± 62.66 [b] | 20.84 ± 0.03 [a] |
|  | N70 | 4343.4 | 3350.0 | 7693.4 | 45,533.77 ± 169.24 [a] | 16.89 ± 0.06 [c] |
|  | CK | 5467.5 | 3200.0 | 8667.5 | 42,238.20 ± 109.25 [b] | 20.52 ± 0.05 [b] |

Note: Lowercase letters are significantly different at the 0.05 level.

## 4. Discussion

Conservation tillage (reduced tillage or no tillage) is known to reduce soil erosion, maintain cropland fertility and improve soil structure, and is one of the most effective options for increasing crop yield and agricultural sustainability [30]. However, prolonged conservation tillage can easily lead to soil compaction, which limits crop root growth and uptake and utilization of subsurface nutrients [31,32]. Henderson [33] studied various crops and found that, compared with conventional tillage, the dry matter quality of each crop under smash ridge tillage was increased by an average of 30%. Therefore, we need to break down the compacted plow bed and maximize the positive impact of conservation tillage on crop productivity. The results of this study show that smash ridge tillage can improve the growth environment of flue-cured tobacco roots and significantly increase the volume of flue-cured tobacco roots under a root system of flue-cured tobacco. The principle should be that smash ridge tillage could increase soil porosity, reduce soil bulk density and penetration resistance, resulting in increased soil ventilation and reduced root growth resistance [34]. At the same time, this study found that smash ridge tillage can increase the maximum leaf area of tobacco and significantly increase the dry matter accumulation, which is similar to the results found by Sun et al. [35], because plant roots could accelerate growth in the improved soil environment and absorb water and nutrients in the deep soil [36], thus improving root vitality and delaying root senescence, maintaining and promoting the supply of nutrients and water to the aboveground parts by the roots, enhancing photosynthesis and respiration of plants, accelerating cell division, increasing nitrate reductase activity and ultimately helping to improve the leaves' area and dry matter weight of tobacco [37,38]. Similar findings appear in Rubio et al. [39] and Colombi and Keller [40], who attributed the increase in crop yield after deep tillage and subsoiling to improvements in water use efficiency, soil nutrients and crops' root growth. The treatment that reduced nitrogen fertilizer by less than 30% did not inhibit growth and development, which may be because smash ridge tillage can reduce nutrient loss, expand the storage space of organic carbon materials [41] and maintain soil fertility compared with conventional tillage [42,43].

Optimizing nitrogen application methods can improve nitrogen use efficiency, and it has been proven that the best way to improve fertilizer use efficiency is to reduce nitrogen application rates [44]. This is because the response of crops to fertilization is strongly affected by soil nitrogen supply, that is, when soil nitrogen content is higher, the response of crop growth to fertilization is reduced [45]. Mao et al. [46] added four levels of N (0, 50,

100 and 150 µg N g$^{-1}$ soil) to the roots of pine and poplar under laboratory conditions to study the effect of soil nitrogen enrichment on the decomposition of fresh roots. The results showed that nitrogen addition was negatively correlated with root decomposition, which indicated that nitrogen addition would not accelerate or even inhibit the root decomposition of poplar, thereby inhibiting the generation of soil nutrients. This study found that the nitrogen content and nitrogen harvest index of smash ridge tillage + reducing nitrogen fertilizer application by 10~30% were higher than those of CK in the middle and late stages of tobacco growth. The growth rate of tobacco plants is lower than that of tobacco plants under conventional farming methods. After the middle and late stages of field growth, because of the developed root system of tobacco plants, the ability to supply nutrients and water to the aboveground parts is stronger, which leads to the gradual acceleration of the growth of the aboveground parts of tobacco plants, which is ultimately better than that of tobacco plants under conventional tillage methods [47]. Cheng et al. [48] found that tillage depth of 15 cm and reducing nitrogen application rate by 15% can increase the root length and root density of maize in the subsoil, and increase the rooting depth, which is beneficial to maintain a higher biomass and nitrogen accumulation.

The results of this study show that smash ridge tillage can increase the yield of flue-cured tobacco and improve the economic benefits, which is similar to the research results of Zheng et al. [49], Chen et al. [50]. However, with the increase in nitrogen fertilizer application after smash ridge tillage, the yield, output value and quality of crops showed a downward trend [51]. Kaur and Arora [52] reported that this is a synergistic effect of water and nitrogen on crops, which means that under the same water level limitation, higher nitrogen content reduces crop biomass and yield, smash ridge tillage can help soil store more water and nitrogen [53] and the sufficient water and nutrients promoted crop growth and ultimate yield, which was in accordance with the general concept that water and nutrient availabilities are important factors in crop production and food security.

## 5. Conclusions

Our results showed that a combination of smash ridge tillage and nitrogen reduction, which mainly resulted in a higher yield through a significantly promoted roots traits and yield components of tobacco (such as leaf area, dry matter weight, nitrogen accumulation), could also reduce production costs and improve economic benefits, with smash ridge tillage + 30% nitrogen reduction being the most effective. Therefore, we believe that the combination of smash ridge tillage and nitrogen reduction is an effective and environmentally friendly way to improve agricultural productivity and nitrogen use efficiency of crops. The results from this study are significant to the development of sustainable and balanced farming in agriculture.

**Author Contributions:** Conceptualization, Z.R.; methodology, Z.R., Y.J., Y.L. and J.L.; software, B.Z., R.H.; validation, Y.Z., J.G. and Q.Z.; formal analysis, B.Z., Y.Z., X.T., Q.Z. and J.G.; investigation, Z.R., J.L. and Y.J.; resources, Y.J., Y.L., Z.X. and F.H.; data curation, Y.J., Y.L., Z.X., J.L. and F.H.; writing—original draft preparation, B.Z., Y.Z.; writing—review and editing, Z.R., J.L. and Y.J.; visualization, Z.R., Y.J. and R.H.; supervision, Y.J., Y.L., Z.X. and Z.R.; project administration, Y.J., Z.X. and Y.L.; funding acquisition, Y.J., Z.X. and R.H. All authors have read and agreed to the published version of the manuscript.

**Funding:** This study was supported by the Key Project of Science and Technology of Hunan Tobacco Company (HN2021KJ05) and the Key Project of Science and Technology of Hunan Tobacco Company Chenzhou Branch (CZYC2021JS10) and Postgraduate Scientific Research Innovation Project of Hunan Province (QL20210166) and Hunan Tobacco Industry Co., Ltd. (202043000934133).

**Data Availability Statement:** The datasets generated and/or analyzed during the current study are available from the corresponding author on reasonable request.

**Conflicts of Interest:** All authors declare that they have no conflict of interest.

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
