# Peer review of "Responses of Tobacco Growth and Development, Nitrogen Use Efficiency, Crop Yield and Economic Benefits to Smash Ridge Tillage and Nitrogen Reduction"

_agronomy, doi:10.3390/agronomy12092097_

Round 1

Reviewer 1 Report

Given the increasing costs of fertilizers, this research work is a good approach to reducing the costs by altering farming techniques such as deep plowing in this case. The manuscript needs some clarifications to the experimental design and formatting. Some of the observations are as below.

The text needs formatting for space, between words, characters, and also the references.

The Abstract has a conflict between past and present tense. Kindly correct the text with such inconsistencies to get a better flow of thoughts.

For example, Line 13: Long term rotary tillage leads to soil layer thinning. This is a known fact as it does seriously limit root growth.

Line 72: Make this into 2 meaningful sentences

Table 1: While the amount of total C, P and K seems to be distributed fairly even in the soil depths by the year, there is a drastic decrease in available P in 2019 at 20-30 cm depth and available K has increased almost 4 times in 2019 especially at 10-20 cm and 20-30 cm depth. Do you have any explanation for this observation and how this has affected your results?

Lines 99 - 102: Experimental design – Why did you not have the same treatments for the 2 years of field work? Except for conventional tillage and fertilization CKa, N100 and N70, there is no scope for comparisons of other N fertilizers (90, 85, 80 and 60) between the years. Kindly explain the logic to this experimental approach. Usually, a minimum of 2 field seasons with the same experimental design is required to validate the results and gives the scope to understand how the weather patterns might also affect the crop. Nutrient acquisition and partitioning also depends on the weather pattern.

The cost comparison for economic value cannot be determined by 1 field season data for N90, 85, 80 and 60. There is something missing in the explanation to connect the dots.

Line 274: and crops’ root growth.

Reviewer 2 Report

Manuscript no 1824769:

Effects of Deep Tillage and Nitrogen Reduction on Growth and Economic Benefit of Tobacco 

The authors Zheng et al. researched the effects of deep tillage towards N reduction on growth and economic benefit of tobacco. They observed the DT improving the roots growth and reduced usage of N fertilizer in a range of 26-30%.

The research is fascinating. However, the manuscript is poorly presented, from the abstract to the conclusion. Some contents are missing; some exist but need to be presented logically and without repetition of the results. The research study should have measured soil status before transplanting tobacco and after harvesting to know the N contents remaining in the soils to the three depths, but it lacked this information. Additionally, I have not seen any interaction effect on tillage depths (0-10, 10-20, 20-30) with N contents. By missing this kind of analysis, it is difficult for a reader to believe if the effect of deep tillage influenced N reduction and growth. I propose the authors need to overhaul the entire manuscript content, including improving the grammar. Some tables are tough to understand their treatments and need great improvement. Significant adjustments are required to make the manuscript easy-readable. My suggestion is to reject the article for publication in Agronomy Journal. For more details, I have highlighted some key issues that need clarification and improvement.

Research Title

Should be improved

Abstract

It needs significant improvement

Introduction

Need to be significantly improved to have a logical flow with a good connection. Some contents are there, while some critical contents are missing and must be included and logically arranged. Literature on how the tillage depths affect soil nutrients needs to be improved. The fonts are of different styles.

Materials and methodologies 

It is unclear if the soil fertility results were done prior to transplanting tobacco seedlings or after harvesting (Table 1). Authors should indicate if samples were collected to measure soil N before and after harvesting each planting season to calculate how the plant efficiently utilized the nutrients uptake. In addition, the section is missing the silt, sand and clay % to prove if the soil textural was sandy.

Experimental design

The treatments were not properly designed as they were not the same for both cropping seasons. According to the authors, in the first season, there were four treatments, while in the second year, there were five treatments. Nevertheless, in Table 2, I noticed that in 2018 there were four treatments, while in 2019, there were six treatments. The clarity of this information is essential. 

Soil analysis in Table 1 was done to the three depths 0-10, 10-20 and 20-30. I wonder why the authors did not collect samples for each depth on soil N to reflect the deep tillage effect on N? 

When was the basal fertilizer applied to the transplanted seedlings, and at what rate per plant? 

Line 113: Some figures for P and K are missing

No details were given for the time of sowing tobacco seedlings and how many grams of tobacco were sown in seedbeds. In addition, seedbed size was not indicated, and transplanting time was not described.

Line 115-120 are full of repeating information with soil terms that need to be corrected. For example, ‘root soil’. Instead, authors should write that “the soil adhered to root….”

Line 115-134: No description of how the tobacco leaf area was calculated

Line 130-133: Equations/formulae should be numbered for quickly referencing the results obtained in due aspects.

Statistical analysis

Line 135: Many details skipped on how the yield analysis and other agronomic traits were analyzed. It must be provided information on what parameters were analyzed through EXCEL and SPSS software and why. The Duncan new complex range methods were used to analyze what? Did the authors use Analysis of Variance (ANOVA) to analyze the data? If yes, was one-way or two ways ANOVA used? Was the interaction analysis based on tillage depth and N nutrients done? 

Results

The authors have not well presented their results and, in most cases, are mixed up with discussion. The authors should have started presenting results on how deep tillage influenced the growth, crop stand, number of leaves, plant height, N levels, etc., logically.

Table 3 is large and needs to be presented in landscape format

Line 203 & 223: Some of the data were not statistically analyzed e.g Table 1, 2, 5, 6 & 7 that make me doubt the presented results. Overall, this section should be critically revised and arranged in good order to help logically discuss results. 

Discussion

It is very poorly presented, and the authors in most cases did not discuss the results but instead discussed literature reviews with full speculations.

In the results and discussion, the wording is awkward and confusing. English language needs improvement.

Conclusion

Need to be focused and should be brief with no repetition of the results and discussion

Reviewer 3 Report

The article entitled “Effects of Deep Tillage and Nitrogen Reduction on
2 Growth and Economic Benefit of Tobacco
is submitted to the Agronomy journal. This study aimed to explore the impact of reducing nitrogen fertilizer after deep tillage on the growth, development and economic properties of tobacco, and explore the reasonable nitrogen fertilizer reduction interval after deep tillage. This paper is a good fit for the journal’s audience and subject matter; specifically It provides a reference for the application of deep tillage
technology in crop production and agricultural sustainable development in the future. The experiment is a field experiment based on tillage.

I recommend this article for publication in the journal with major revisions, based on the comments and questions detailed below. I also uploaded a pdf file for minor revision .  

1.     Revise the manuscript to improve its writing flow. Like the 1st sentence of the introduction, part is not linked with the 2nd sentence.

2.      The manuscript has extra or no spacing between two sentences or words, which I have highlighted in the pdf file. Revise throughout the MS.

3.      Line 39-41. If the sentence is difficult to understand, revise it.

4.      All the citation style is not according to the journal format. The authors should read the Authors' guidelines first before submitting the paper.

5.      There are grammatical mistakes in the text, authors must remove it.

6.      Line 53. “Deep tillage can deepen the tillage depth”. What about the nutrient leaching?

7.      Line 62. Hm2 is not a standard unit, convert it to ha-1

8.      Line 63. Replace with "To improve crop growth and production, farmers increased the rate of nitrogen fertilizer application in common tillage farming system in China, which is not environmental friendly.

9.      Line 65-74. If the sentence is too long, add references and separate with parts.

10.   Line 75 to 80. Write clear objectives of the study.

11.   Revise the experimental design and make it readable.

12.  Why did you change the experimental treatments during 2019? What were your main objectives?

13.   Line 115. Which method? Manual or using any scanning machine?

14.   Give numbers to titles and subtitles.

15.   Conclusion should be on the basis of your results. First, conclude your results the suggest the methods to others.  

Round 2

Reviewer 2 Report

Authors have improved the articles. However, they need to recheck some figures in Table 5 and 6 as some have no significant letters. 

Author Response

Response  to reviewer 2's Comments

Point 1: Authors have improved the articles. However, they need to recheck some figures in Table 5 and 6 as some have no significant letters. 

Response 1: Thanks for pointing this out. We have rechecked and corrected the figures in Table 5 and 6.

Reviewer 3 Report

The authors revised the manuscript accordingly to my suggestions and comments. However, the manuscript still needs improvement. The flow of the text is not readable. I have suggested

Check the author’s name “and” is located in the last.

Have you used Smash ridge tillage? If yes, then replace the deep tillage word with Smash ridge tillage.

Line 20 and 25 ha-1 replace with ha-1 correct it throughout the manuscript

Line 33 This is not enough for the soil background

Line 34, How traditional farming methods are seriously damaging the soil quality of cultivated land? Give an example in the same sentence.

Line 52. Add reference https://doi.org/10.1016/j.sjbs.2020.11.054

Line 62 Replace “which is not environmentally eco-friendly.

Line 69 Farming methods or tillage methods? Because farming includes several methods, i.e., pesticide, irrigation, tillage, sowing methods, etc

Line 141-142. Write the formula in one line

Line 251 and after that. Text format is not the same

Table 2; replace g/kg with g kg-1 and throughout the MS

Write detail about a and b with treatments names in the table caption, some treatments have a and b while some don’t have why?

Table 6( g/per plant)replace with g plant-1 and throughout the MS, for example, g/per leaf

The manuscript still needs improvement, especially the text format and units.

Line 265-266 provide valid reasons for improving in leaf area,

Line 299-306, the discussion for an increase in yield is not enough. Why the yield was increased, give a valid reason/explanation from your results.

The conclusion needs improvement. Accordingly, in your own words,

The primary mechanism of deep tillage is improvement in roots attributes; if roots are better, they can increase nutrient uptake. While more N uptake can improve leaf area and dry matter production. These all increments improve tobacco yield.  
